# Multivariable two-sample Mendelian randomization estimates of the effects of intelligence and education on health

**Neil Martin Davies[1,2]\*, W David Hill[3,4], Emma L Anderson[1,2], Eleanor Sanderson[1,2], Ian J Deary[3,4], George Davey Smith[1,2]**

[1]Medical Research Council Integrative Epidemiology Unit, University of Bristol, Bristol, United Kingdom; [2]Bristol Medical School, University of Bristol, Bristol, United Kingdom; [3]Centre for Cognitive Ageing and Cognitive Epidemiology, University of Edinburgh, Edinburgh, United Kingdom; [4]Department of Psychology, University of Edinburgh, Edinburgh, United Kingdom

**Abstract** Intelligence and education are predictive of better physical and mental health, socioeconomic position (SEP), and longevity. However, these associations are insufficient to prove that intelligence and/or education cause these outcomes. Intelligence and education are phenotypically and genetically correlated, which makes it difficult to elucidate causal relationships. We used univariate and multivariable Mendelian randomization to estimate the total and direct effects of intelligence and educational attainment on mental and physical health, measures of socioeconomic position, and longevity. Both intelligence and education had beneficial total effects. Higher intelligence had positive direct effects on income and alcohol consumption, and negative direct effects on moderate and vigorous physical activity. Higher educational attainment had positive direct effects on income, alcohol consumption, and vigorous physical activity, and negative direct effects on smoking, BMI and sedentary behaviour. If the Mendelian randomization assumptions hold, these findings suggest that both intelligence and education affect health.
DOI: https://doi.org/10.7554/eLife.43990.001

\*For correspondence:
neil.davies@bristol.ac.uk

## Introduction

Intelligence and educational attainment are associated with many socioeconomic and health outcomes (*Cutler and Lleras-Muney, 2006*; *Clark and Royer, 2013*; *Deary and Johnson, 2010*; *Deary, 2012*; *Hill et al., 2016a*). However, the causal relationships between intelligence, education, and health outcomes are unclear, in part, because intelligence and education are strongly correlated. On average, children who score more highly in intelligence tests tend to remain in school for longer, (*Deary et al., 2007*) and people who remained in school for longer tend to have higher intelligence later in life (*Ritchie and Tucker-Drob, 2018*). Intelligence and educational attainment are partially heritable, and strongly genetically correlated ($r_g$=0.70) (*Lee et al., 2018*; *Hill et al., 2019*; *Hill et al., 2016b*; *Hill et al., 2018*). A review of the effects of educational attainment on socio-economic and health outcomes in later life found some evidence that increases in education led to lower mortality, especially for older cohorts (*Galama et al., 2018*). However, there was little consistent evidence of effects on other outcomes such as obesity (*Galama et al., 2018*). A systematic review of 28 studies provided evidence using quasi-experiential designs that each year of education causes measured IQ to increase by on average 1 to 5 points (*Ritchie and Tucker-Drob, 2018*). Large prospective studies have found that intelligence test scores taken at age 11 strongly associate with educational attainment later in life, a link that is, in part, explained by shared genetic effects (*Deary et al., 2007*; *Calvin et al., 2012*). However, there are no quasi-experimental studies of the effects of intelligence

**eLife digest** Highly educated people tend to be healthier and have higher incomes than those with less schooling. This might be because education helps people adopt a healthier lifestyle, as well as qualifying them for better-paid jobs. But, on average, highly educated people also score more highly on cognitive tests. This may explain why they tend to adopt healthier behaviours, such as being less likely to smoke. Because education and intelligence are so closely related, it is difficult to tease apart their roles in people's health.

Davies et al. have now turned to genetics to explore this question, focusing on genetic variation associated with intelligence and education levels. Analysing genetic and lifestyle data from almost 140,000 healthy middle-aged volunteers from the UK Biobank study suggested that together, intelligence and education influence many life outcomes, but also that they have independent effects. For instance, there is evidence that more intelligent people tend to earn more, irrespective of their education. However, more educated people also tend to earn more, even after accounting for their intelligence. They also tend to have lower BMIs, be less likely to smoke, and engage in less sedentary behaviour and more frequent vigorous exercise in midlife. For each of these outcomes, the effects of education are all in addition to the effects of intelligence.

Education and intelligence thus affect life outcomes together and independently. Overall, the results of Davies et al. suggest that extending education, for example by increasing school-leaving age, could make the population as a whole healthier. However, the individuals in the current study grew up when smoking was far more common than it is today. Some of the observed effects on health may thus be due to differences in smoking rates between groups with different levels of education. If so, increasing education may not have as much impact today as it did in the past. It is also possible that these findings reflect the effects of the family environment, for example how parents influence their offspring. Larger studies are needed to investigate this hypothesis.

DOI: https://doi.org/10.7554/eLife.43990.002

on education because intelligence is less subject to perturbation by natural experiments such as policy reforms. In order to design successful interventions aimed at the amelioration of health conditions, it is necessary to determine the extent to which intelligence, education, or both, are causal factors for health outcomes. If the health differences are mainly due to differences in intelligence that are independent from education, then changes to the length of schooling are unlikely to affect population health. If, however, educational attainment has a causal effect on health and socioeconomic outcomes later in life, and these effects are independent of intelligence, the health of the population may be able to be raised by implementing a policy change aimed at increasing the duration of education (*Deary and Johnson, 2010*).

Mendelian randomization is an approach that can provide evidence about the relative causal effects of intelligence and education on social and health outcomes under specific assumptions (*Davey Smith and Ebrahim, 2003*; *Davies et al., 2018a*). Mendelian randomization generally uses single nucleotide polymorphisms (SNPs) that associate with traits of interest, in this case intelligence and educational attainment, as proxy variables for a trait. At each SNP, offspring inherit at random one of their mother's two possible alleles and one of their father's two possible alleles. With the exception of somatic mutation, SNPs are invariant post-conception, so it is not possible for the environment or developing disease to affect inherited DNA. Thus, SNPs are not affected by reverse causation, which can distort causal interpretations of observational epidemiological associations. Segregation of alleles at germ cell formation (Mendel's first law) and the independent assortment of alleles with respect to the rest of the genome excepting proximal DNA segments (Mendel's second law), and the lack of environmentally influenced effects on survival from conception through to live birth and study entry, leads to genetic variants being largely unrelated to factors that would confound conventional observational studies (*Davey Smith, 2011*).

Instrumental variable interpretations of Mendelian randomization analysis depend on three assumptions, (1) the genetic variants associate with the risk factors of interest, (2) the genetic variants-outcome associations are not confounded by potentially unmeasured factors, and (3) the genetic variants only affect the outcomes via their effect on the exposures of interest (in this case

intelligence or education) (*Davies et al., 2018a*). One approach would be to estimate the effects of intelligence and education separately using SNPs identified in GWAS for intelligence or education (*Hill et al., 2019*; *Okbay et al., 2016*). However, this could be misleading, as SNPs that affect intelligence are likely to also affect education and vice versa. Thus, the SNPs associated with intelligence and education are pleiotropic – and most affect both traits. However, it is unclear whether any effects of the SNPs on the health and socioeconomic outcomes later in life via education and intelligence are vertically or horizontally pleiotropic (*Davey Smith and Hemani, 2014*).

Consider a Mendelian randomization study with one exposure - education. Vertically pleiotropic effects would occur if the SNPs being used as proxies for education affect the outcomes first via their effects on intelligence, which then has a downstream effect on educational attainment (*Figure 1a*). Horizontally pleiotropic effects could occur if a SNP being used as a proxy for education directly affected an outcome via intelligence without being mediated via education (*Figure 1b*). In a Mendelian randomization analysis using education as a single exposure, this horizontal pleiotropy would violate the third Mendelian randomization assumption. Alternatively, the SNPs being used as proxies for education could affect intelligence, and consequently educational attainment, but all of the effects of the SNPs on the outcome could be mediated via intelligence (*Figure 1c*). Pleiotropy would have similar consequences for a Mendelian randomization study with intelligence as a single exposure. One method which can provide evidence about which of these scenarios is most likely is multivariable Mendelian randomization (*Sanderson et al., 2018*; *Burgess et al., 2015*). This approach simultaneously estimates the effects of two or more risk factors using a potentially overlapping set of SNPs. Multivariable Mendelian randomization estimates the direct effect of each risk factor – i.e. the direct effect of intelligence on outcomes that is not mediated via the effect of intelligence on education, and the direct effect of education that is not mediated via the effect of education on intelligence (*Figure 2*). It is important to note that multivariable Mendelian randomization does not overcome bias due to horizontally pleiotropic effects via other mechanisms, for example, effects via character or personality.

Here, we used a large sample from the UK Biobank and SNPs associated with intelligence and education to estimate the direct effects of intelligence and education on a range of health and socioeconomic outcomes. Our primary analysis uses two-sample multivariable Mendelian randomization of the effects of intelligence and education on a range of socioeconomic and health outcomes. This approach makes most efficient use of available data. We present single sample Mendelian randomization analysis using polygenic risk scores in the UK Biobank as sensitivity analyses.

## Results

The characteristics of 138,670 participants of UK Biobank who met our quality control and inclusion criteria for our primary two-sample multivariable analysis are described in *Table 1*. We take estimates of the SNP-intelligence and SNP-education associations from published GWAS of intelligence and education (Hill et al. and Okbay et al.) and estimates of the SNP- outcome associations using participants of the UK Biobank who were not included in either GWAS. The intelligence and education GWAS used overlapping samples, but do not overlap with the sample used to estimate the SNP-outcome associations. See *Supplementary file 1 - Figure 1* for a flow chart of participants' inclusion and exclusion into the analytic samples. The sample was 45.5% male and on average were 57.3 years old when they visited the assessment centre. The prevalence of the clinical outcomes in this sample ranged from 1.5% for stroke to 34.6% for a broad measure of depression. Of the participants in this sample, 2.0% died by the end of linked data follow-up. A total of 45.2% of the sample had ever smoked, and 9.2% were current smokers.

### Bidirectional Mendelian randomization of intelligence and education

We investigated the direction of causation between intelligence and education using bidirectional two-sample Mendelian randomization. We restricted analysis to SNPs that were available in the *Hill et al. (2019)* intelligence genome-wide association study (GWAS), the discovery sample of *Okbay et al. (2016)* educational attainment GWAS, and the UK Biobank Haplotype Reference Consortium panel (HRC). We selected all SNPs that were associated with intelligence at $p < 5 \times 10^{-08}$. Of these SNPs, we then selected the lead SNPs within each 10,000 kb genomic region by selecting the SNP with the lowest p-value. We then repeated this process to select all SNPs that associated with

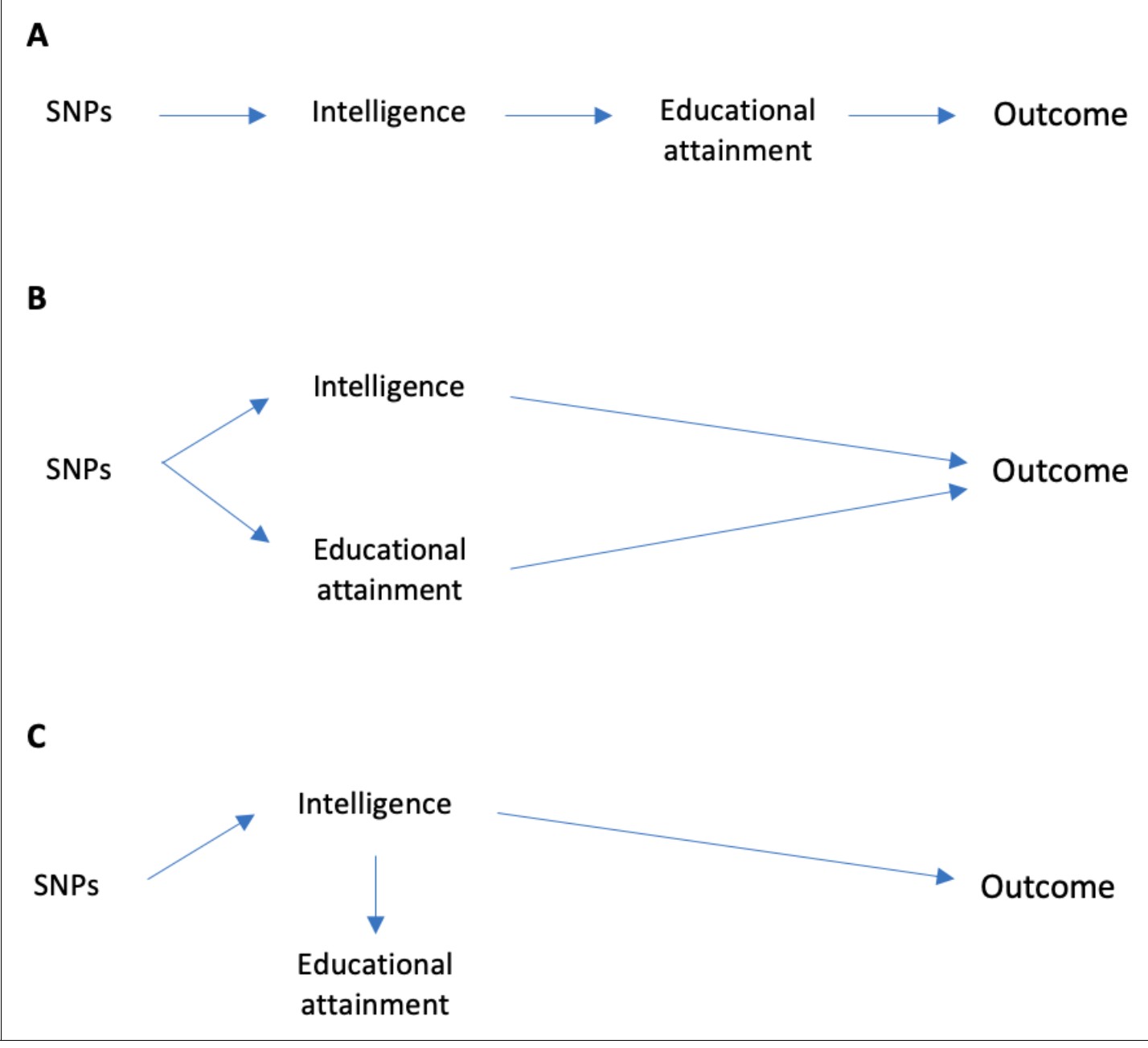

**Figure 1.** Possible explanations for associations of SNPs and intelligence, education and outcomes later in life. (**A**) Vertical pleiotropy: SNPs associated with intelligence and educational attainment could be vertically pleiotropic. This could occur if all of the effect of the SNPs on the outcomes is mediated via their effects on intelligence, the effect of intelligence on educational attainment, and the effect of educational attainment on the outcome. Confounders omitted from this figure for clarity. (**B**) Horizontal pleiotropy: The SNPs could be associated with the outcome via intelligence and educational attainment because of horizontal pleiotropy. This would occur if the SNPs had effects on the outcome via intelligence or education that were not mediated via the other trait. (**C**) Confounding pleiotropy: The SNPs could be associated with the outcomes, intelligence and educational attainment because of the effect of the SNPs on intelligence, and hence education, but that education had no direct effect on the outcome (i.e. the effects of the SNPs were entirely mediated via intelligence. This would occur if the SNPs had effects on the outcome via intelligence or education that were not mediated via the other trait (or vice versa). Multivariable Mendelian randomization does not overcome bias due to other pleiotropic effects by pathways other than intelligence or education.

DOI: https://doi.org/10.7554/eLife.43990.003

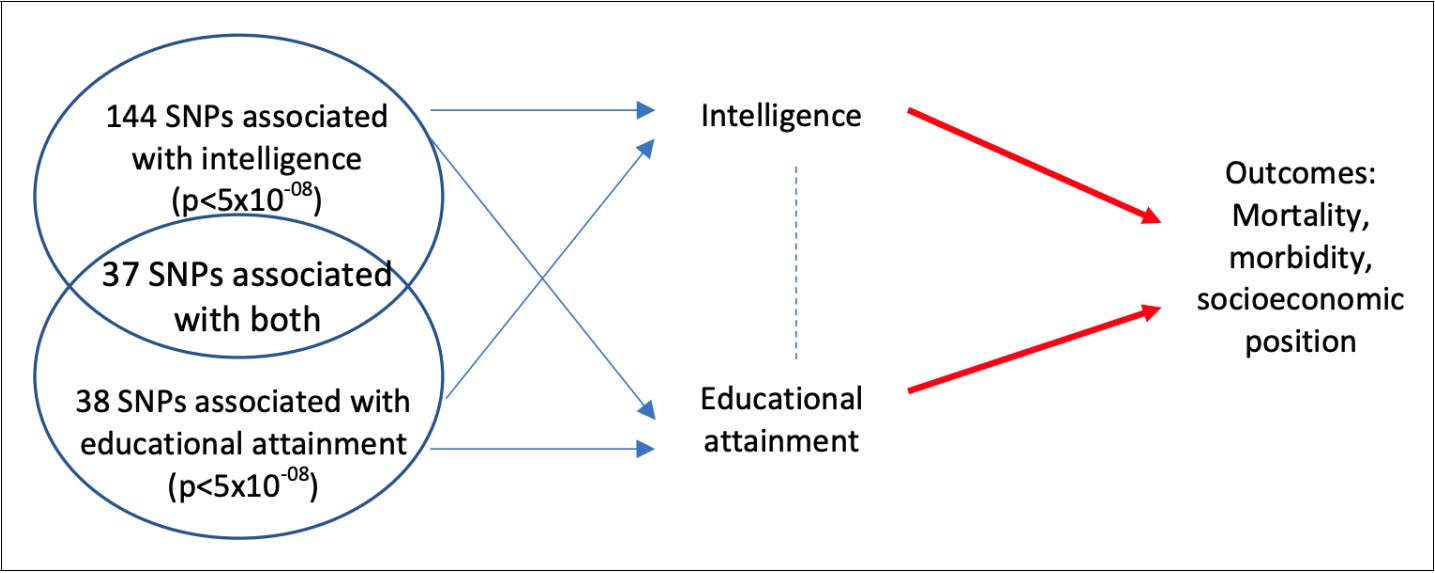

**Figure 2.** Multivariable Mendelian randomization estimates the direct effect of intelligence and education on the outcomes. The direct effect (red arrow) excludes any effect of either intelligence (or education) that is mediated via education (intelligence) on the outcome. It requires genetic variation that explains a sufficient proportion of the variation in intelligence and education conditional on the other trait (*Davey Smith and Hemani, 2014*). It uses a set of SNPs that associate with intelligence and/or education at p<5×10$^{-08}$.

DOI: https://doi.org/10.7554/eLife.43990.004

education at $p<5 \times 10^{-08}$, and selected lead SNPs within each region. This resulted in 270 SNPs that were associated with either intelligence or education, or both. Some of these SNPs represented the same signal, so we clumped a combined list of SNPs by selecting the SNP with the lowest p-value for education. This resulted in 219 SNPs, of which 144 were associated with intelligence, but not education at $p<5 \times 10^{-08}$, 38 were associated with education, but not intelligence at $p<5 \times 10^{-08}$, and 37 were associated with both intelligence and education at $p<5 \times 10^{-08}$. All 219 SNPs associated with either trait were included in the multivariable Mendelian randomization analysis for both intelligence and education (for a flow chart of selection of SNPs see *Supplementary file 1 - Figure 2*).

We estimated the effects of intelligence on education and vice versa using non-overlapping data from the UK Biobank. We used two-sample summary data Mendelian randomization methods including the inverse variance weighted (IVW), MR-Egger, weighted median and modal estimators, see methods for details (*Bowden et al., 2016a*; *Bowden et al., 2016a*; *Hartwig et al., 2017*). The inverse variance weighted (IVW) estimates implied that a one standard deviation (SD) increase in intelligence increased years of education by 0.52 SD (95% CI: 0.48 to 0.56, p-value=2.2 × 10$^{-145}$). The IVW estimates imply that a one SD increase in education increased intelligence by 0.77 SD (95% CI: 0.68 to 0.86, p-value=3.0 × 10$^{-62}$). There was more evidence of heterogeneity in the estimated effects of education across the SNPs ($I^2$ = 0.60, 95% CI: 0.48 to 0.69) than in the estimated effects of intelligence ($I^2$ = 0.48, 95% CI: 0.38 to 0.56). The heterogeneity statistic indicates the variability of the estimated effects between SNPs; a value of zero indicates no heterogeneity and one indicates high heterogeneity. One explanation for this heterogeneity across SNPs is if the SNPs have horizontally pleiotropic effects. The effects of each of the SNPs on intelligence and education, along with the IVW, MR-Egger, weighted median and weighted mode estimates are presented in *Figure 3* and *Supplementary file 1 - Table 1 and 2*. The estimated effect of intelligence was consistent and robust across four different estimators (IVW, MR-Egger, weighted median and weighted mode). The estimated effect of education was robust across IVW, weighted median and weighted mode, but the MR-Egger estimate was attenuated by over half, one explanation for this is if the education SNPs have unbalanced horizontally pleiotropic effects on intelligence or the INSiDE assumption is violated. There was little statistical evidence that the estimated effects of intelligence on education were biased by horizontal pleiotropy (MR-Egger intercept = 0.001, 95% CI: −0.005 to 0.003,

**Table 1.** Characteristics of 138,670 participants of the UK Biobank used to estimate the association of the SNPs and the outcomes. The sample is more educated and healthier than the UK population. The sample was restricted to participants included in the two-sample analysis used for our primary results.

| Covariates | | Mean/percent | Standard deviation/ count |
|---|---|---|---|
| Male | 138,670 | 45.5% | 63,111 |
| Year of birth | 138,670 | 1951 | 8.08 |
| Age at assessment centre visit | 138,670 | 57.3 | 8.08 |
| Exposures | | | |
| Intelligence* | 137,396 | 6.20 | 2.10 |
| Educational attainment | 137,354 | 14.55 | 5.16 |
| Outcomes | | | |
| Hypertension | 134,751 | 24.9% | 33,494 |
| Diabetes | 137,883 | 4.3% | 5988 |
| Stroke | 138,417 | 1.5% | 2090 |
| Heart attack | 138,417 | 2.4% | 3260 |
| Depression | 137,733 | 34.6% | 47,624 |
| Cancer | 138,078 | 13.4% | 18,482 |
| Mortality | 138,670 | 2.0% | 2783 |
| Ever smoker | 138,044 | 45.2% | 62,365 |
| Smoker | 138,044 | 9.3% | 12,831 |
| Income over £18 k | 117,750 | 76.9% | 90,537 |
| Income over £31 k | 117,750 | 51.0% | 60,077 |
| Income over £52 k | 117,750 | 24.9% | 29,358 |
| Income over £100 k | 117,750 | 5.0% | 5871 |
| Grip strength (kg)* | 138,432 | 7.8% | 11.04 |
| Height (cm)* | 138,374 | 168.74 | 9.27 |
| BMI (kg/m2)* | 138,241 | 27.36 | 4.72 |
| Diastolic blood pressure (mmHg)* | 132,955 | 82.31 | 10.16 |
| Systolic blood pressure (mmHg)* | 132,954 | 138.24 | 18.64 |
| Alcohol consumption (one low, five high)* | 138,550 | 3.15 | 1.48 |
| Hours of television viewing per day* | 133,714 | 2.88 | 1.62 |
| Vigorous physical activity (days/week)* | 131,529 | 1.82 | 1.95 |
| Moderate physical activity (days/week)* | 131,561 | 3.60 | 2.33 |

Notes: * Intelligence used in the single sample analysis reported in Supplementary Table 4. Both intelligence and educational attainment were normalised mean zero, standard deviation one for these analyses.

DOI: https://doi.org/10.7554/eLife.43990.005

p-value=0.68). However, these tests for pleiotropy are likely to have low power. There was modest evidence that the estimated effects of education on intelligence were affected by horizontal pleiotropy (MR-Egger intercept = 0.009, 95% CI: 0.001 to 0.016, p-value=0.03).

## Univariable Mendelian randomization

We estimated the total effect of intelligence and education on each of the social and health outcomes using two-sample summary data Mendelian randomization (*Supplementary file 1 - Figure 3*). This approach estimates the total effect of a one SD change in intelligence or education including any effects mediated through the other (or any other, for example character) trait. The estimates of the total effects for each outcome for intelligence and education were generally in a consistent direction. The inverse-variance weighted estimates imply that a one standard deviation increase in

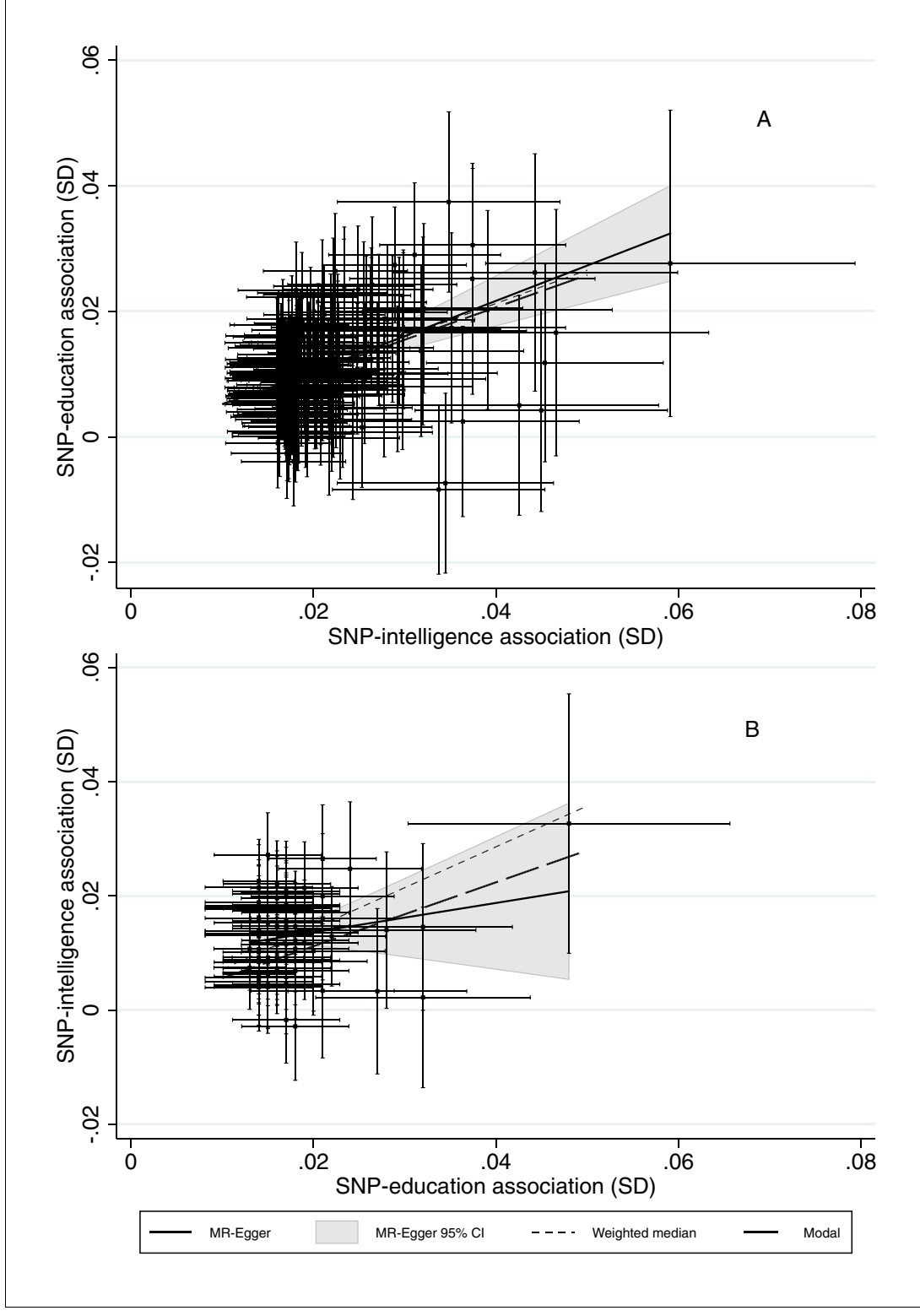

**Figure 3.** The bidirectional effects of SD differences in years of education and intelligence in UK Biobank. The error bars indicate 95% confidence intervals around the estimated phenotype-SNP associations. Two sample multivariable Mendelian randomization using results from *Hill et al. (2019)*, *Okbay et al. (2016)* and UK Biobank data. These results suggest that intelligence increases length of schooling and that higher education leads to higher intelligence. Pleiotropy robust methods, such as MR-Egger suggested little bias in the IVW estimates of the effect of intelligence on education, but that the IVW results may substantially overestimate the effect of education

*Figure 3 continued on next page*

*Figure 3 continued*

on intelligence. Estimates adjusted for month of birth, year of birth, sex, interaction of sex and year of birth and the first 40 principal components. For panel A, Q-stat = 368.7 (p<0.0001), $I^2_{gx}$=34.7%. For panel B, Q = 183.3 (p<0.0001), $I^2_{gx}$=12.5%.

DOI: https://doi.org/10.7554/eLife.43990.006

intelligence reduced risk of: high blood pressure by 4.0 percentage point (pp) (95% CI: 0.02 to 0.06, p-value=$1.2\times10^{-05}$); diabetes by 0.8pp (95% CI: 0.1 to 1.6, p-value=0.04); having had a heart attack by 1.2pp (95% CI: 0.7 to 1.7, p-value=$5.5 \times 10^{-07}$); of reporting having seen a GP for nerves, anxiety, tension or depression by 0.5pp (95% CI: 0.3 to 0.5, p-value=$8.1\times10^{-08}$). The results suggested that a one standard deviation increase in intelligence was unlikely to have a large effect on risk of mortality or cancer (risk difference = $-0.001$, 95% CI: $-0.006$ to 0.003, p-value=0.57, and = $-0.01$, 95% CI: $-0.02$ to 0.003, p-value=0.12). A one standard deviation reduced risk of being an ever or current smoker by 4.9pp (95% CI: 2.8 to 6.9, p-value=$4.5 \times 10^{-06}$) and 3.5pp (95% CI: 2.4 to 4.7, p-value=$1.2\times10^{-09}$). Intelligence also had substantial total effects on household income, increasing the probability of reporting a household income higher than £18,000, £31,000, £52,000 and £100,000 by 11.4pp (95% CI: 9.8 to 12.9, p-value<$1.4\times10^{-45}$), 13.4pp (95% CI: 11.5 to 15.3, p-value=$2.1\times10^{-44}$), 11.9pp (95% CI: 10.3 to 13.6, p-value=$3.4\times10^{-47}$), and 4.1pp (95% CI: 3.3 to 4.9, p-value=$7.7\times10^{-24}$) respectively. Higher intelligence generally had beneficial total effects on all of the continuous outcomes. A one standard deviation increase in intelligence was estimated to increase: grip strength by 0.32 kg (95% CI: 0.03 to 0.67, p-value=0.07); height by 1.44 cm (95% CI: 0.95 to 1.94, p-value=$1.4\times10^{-08}$); alcohol consumption by 0.29 (95%CI: 0.23 to 0.35); days of vigorous physical activity per week by 0.17 (95%CI: 0.10 to 0.24, p-value=$6.7\times10^{-06}$); and moderate physical activity by 0.32 (95%CI: 0.23 to 0.40, p-value=$1.1\times10^{-13}$) and reduce: BMI by 0.91 kg/m$^2$ (95% CI: 0.65 to 1.18, p-value=$7.7\times10^{-12}$); and systolic and diastolic blood pressure by 1.09 (95%CI: 0.65 to 1.54, p-value=$1.5 \times 10^{-06}$) and 1.86 (95%CI: 1.12 to 2.61, p-value=$9.1\times10^{-07}$) mmHg respectively; and hours watching television per day by 0.49 (95%CI: 0.42 to 0.57, p-value=$4.3\times10^{-20}$).

The effects of a one SD increase in education were very similar in direction and magnitude to those of intelligence. An exception was that there was little evidence that education affected frequency of vigorous physical activity (mean difference = 0.00 95% CI: $-0.15$ to 0.15, p-value=0.997).

## Multivariable Mendelian randomization

Next, we estimated the direct effect of each exposure using multivariable Mendelian randomization (*Sanderson et al., 2018*). The direct effect of intelligence (or, *mutatis mutandis*, education) is the effect of intelligence that is not mediated via education (or intelligence). Multivariable Mendelian randomization estimates the effects of two exposures using the two sets of (overlapping) SNPs as instruments. We restricted the analysis to SNPs in linkage equilibrium which were identified in the intelligence and/or education GWAS at p<$5 \times 10^{-08}$ clumped on r2 = 0.01 within 10,000 kb using the 1000 genomes reference panel (*Hemani et al., 2018*). Some SNPs that were selected from the intelligence and education GWAS were closely positioned in the genome and were correlated. For these pairs of SNPs, we selected the SNP that most strongly associated with education in the GWAS. Sensitivity analysis which clumped SNPs using their association in the intelligence GWAS is presented in the supplementary materials; however, the results were virtually identical. The instruments strongly predicted both education and intelligence in the single sample analysis (the minimum Sanderson-Windmeijer multivariable F-statistic was 21.6) (*Sanderson et al., 2018*; *Sanderson and Windmeijer, 2015*). Sanderson-Windmeijer F-statistics tests the strength of the SNP-exposure conditional on the other exposure (intelligence or education). Because the effects of the SNPs on intelligence and education are similar (but not identical), the Sanderson-Windmeijer multivariable F-statistics are smaller than standard univariable F-statistics. The estimates of the direct effects presented in *Figure 4* are less precise than the estimates of the total effects. It is not possible to test the strength of the instruments to jointly predict both of the exposures for the two-sample analysis. However, the Sanderson-Windmeijer tests of the strength of the instruments in the single sample analysis is likely to provide a lower bound of their strength.

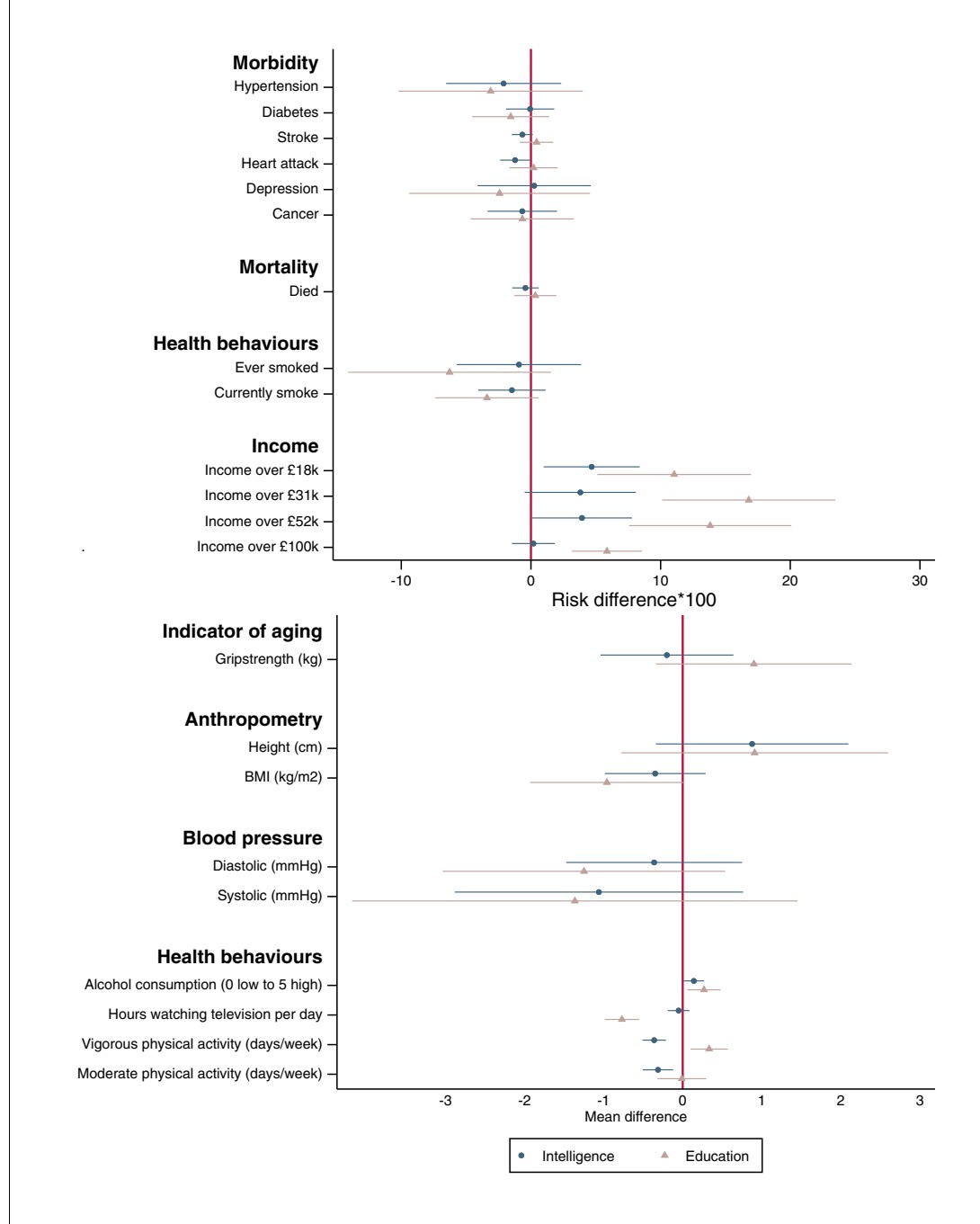

**Figure 4.** The direct effects of SD changes in years of education and intelligence on later outcomes in UK Biobank. The error bars indicate 95% confidence intervals around the estimated effects. Estimated using two sample multivariable Mendelian randomization. Higher intelligence had direct effects on higher household income and alcohol consumption and less moderate and vigorous physical activity. Higher education had direct effects on decreased smoking, BMI, and sedentary behaviour and increased household income and rates of vigorous physical activity. These are estimates of the direct effects of intelligence (education) that are not mediated via education (intelligence).

DOI: https://doi.org/10.7554/eLife.43990.007

## The direct effects of intelligence

In the two-sample analysis the estimates of the direct effects of intelligence were attenuated compared to the total effects. As seen (*Figure 4* and *Supplementary file 1 - Figure 3*), a one SD increase in intelligence score increased the probability of a household income above £18,000 and

£52,000 by 5.2pp (95% CI: 1.5 to 8.9, p-value=0.007) and 4.7pp (95% CI: 0.8 to 8.6, p-value=0.02) respectively. A SD increase in intelligence score increased alcohol consumption by 0.19 (95%CI: 0.06 to 0.32, p-value=0.005 on a five-unit Likert scale). A SD increase in intelligence score decreased rates of vigorous and moderate physical activity by 0.34 (95%CI: 19.6 to 49.1, p-value=$8.2 \times 10^{-06}$) and 0.31 (95%CI: 0.12 to 0.50, p-value=$1.9 \times 10^{-03}$) days per week.

### The direct effects of education

The direct effects of education that were attenuated compared to the total effects. A one SD increase in education increased the probability of having higher income across the entire income distribution. The direct effects of education on household income on the risk difference scale were between 2.0 and 6.6 times as large as the direct effects of intelligence. A SD increase in education resulted in a 1.00 kg/m$^2$ (95% CI: 0.06 to 1.93, p-value=0.04) decrease in BMI, which was larger than the direct effects of intelligence. The direct effects of education on alcohol consumption were similar to intelligence: an increase of 0.21 (95%CI: 0.01 to 0.41, p-value=0.04). Each SD increase in education reduced television consumption by 46.7 min per day (95% CI: 33.6 to 59.8, p-value=$1.8 \times 10^{-11}$) and increased vigorous physical activity by 0.31 days per week (95% CI: 0.09 to 0.54, p-value=0.007).

### Sensitivity analysis

#### Individual-level allele score approach

We conducted a sensitivity analysis using individual-level data from the UK Biobank in the sample of 93,135 participants who took the verbal-numeric reasoning test who were not in the interim release. We identified 16 independent SNPs that were associated with intelligence in the *Sniekers et al. (2017)* GWAS, which used only the interim release of the UK Biobank. We selected SNPs associated with education using the discovery sample of *Okbay et al. (2016)*, which did not use UK Biobank. This approach ensured that we had no sample overlap. We used fewer SNPs for intelligence, and the effects of these SNPs were less precisely estimated; therefore, the SNPs explained less of the variance in intelligence and education than in the primary two-sample analysis (*Supplementary file 1 - Table 4*). As a result, the standard errors for the effects of intelligence and education were between 1.87 and 5.09 times as large as for the two-sample analysis. The effects were similar in magnitude to the two-sample analysis. There was little evidence of substantial direct effects of intelligence on any outcome except negative effects on moderate and vigorous physical activity. In addition to the effects detected in the two-sample analysis, we found evidence of direct effects of education on risk of stroke, ever and current smoking, height and moderate physical activity. Hausman-Durbin-Wu tests suggested substantial evidence of heterogeneity between the instrumental variable and the ordinary least squares estimates (*Hausman, 1978*).

We repeated the individual level analysis with 486 independent SNPs associated with educational attainment by Lee et al. and the Sniekers et al. intelligence score. Sniekers et al. used 55,000 samples from the UK Biobank. Therefore, these samples overlap, so may be affected by weak instrument bias. We constructed unweighted scores which will reduce the risk of weak instrument bias which can potentially affect weighted allele score analyses with overlapping samples (*Burgess et al., 2016*). The results are in *Supplementary file 1 - Table 5*. The results were between 50.0% and 70.9% more precise than the single sample results using the Sniekers et al. score. They had larger values of the Sanderson-Windmeijer F-statistics. Overall, these results were consistent in direction with the primary analysis.

We investigated whether the genetic variants associate with age at baseline. Without adjustment a one standard deviation increases in the Sniekers et al. and Okbay et al. scores were associated with 0.05 (95%CI: 0.0012 to 0.098, p-value=0.044) and 0.058 (95%CI: 0.020 to 0.096 p-value=0.003) increases in age in years at baseline. After adjustment for sex and the principal components these differences fell to 0.041 (95%CI: −0.007 to 0.089, p-value=0.095) and. 0580 (95%CI: −0.0195 to 0.096 p-value=0.003) respectively. We found little evidence that the results were affected by including a broader set of covariates (*Supplementary file 1 - Figure 5*). Finally, we used bias component plots to investigate the relative bias in the multivariable adjusted and instrumental variable analysis (*Davies et al., 2017*). This approach compares the omitted variable bias that occurs if a specific measured covariate is omitted from the multivariable adjusted and instrumental variable regression.

We found some evidence that the scores, particularly the education score, associated with longitude and latitude of place of birth and distance from London (*Supplementary file 1 - Figure 6 and 7*). We investigated whether the single sample results were due to dynastic effects or assortative mating using a sample of siblings and including a family fixed effect. There was little evidence that these results differed from the bivariate two-sample Mendelian randomization (*Supplementary file 1 - Table 7*). However, the sample of siblings was relatively small, and these results may be due to lack of power. Overall this suggests that while the there is some evidence that the scores associate with covariates, there is little evidence from these sensitivity analyses that these violations of the Mendelian randomization assumptions affects our results.

## Discussion

Education and intelligence are strongly phenotypically and genetically correlated and are associated with many outcomes across the life course. However, the direction of causation underlying these relationships is not clear, and nor is the direction of association between intelligence and education. In this study, we used univariate and multivariable Mendelian randomization to estimate the total and direct effects of both years of education and intelligence on health and social outcomes. We used SNPs associated with intelligence or education, or both, at $p < 5 \times 10^{-08}$ to estimate the total and direct effects on a range of outcomes. The estimated total effects of intelligence indicate the overall effects including effects of intelligence mediated via education. The estimated total effects of education indicate the overall effects of education including the effects mediated via intelligence. Whereas the direct effects of intelligence are those that are not mediated via education, and the direct effects of education are those that are not mediated via intelligence. Multivariable Mendelian randomization does not overcome bias due to other pleiotropic effects by pathways other than intelligence or education. Both intelligence and educational attainment had beneficial total effects on most outcomes (the exception being alcohol consumption). Intelligence had positive direct effects on income and alcohol consumption, and negative direct effects on moderate and vigorous physical activity. Education had positive direct effects on household income, BMI, alcohol consumption, and rates of vigorous physical activity, and negative direct effects on sedentary behaviour. These estimates suggest that both intelligence and education affect a range of socioeconomic and health-related outcomes. Many of the direct effects for both traits were substantially attenuated compared to the total effects. For example, the direct effects of intelligence on income were 54% to 89% smaller than the total effect. The direct effects of education were 27% to 38% smaller than the total effects.

Previous studies have demonstrated that both intelligence and education positively associate with health and longevity (*Hill et al., 2019*; *Batty et al., 2007*; *Gottfredson and Deary, 2004*; *Plomin and Deary, 2015*; *Davey Smith et al., 1998*). A systematic review of cohort studies found people who had a one standard deviation higher childhood intelligence had 24% (95%CI: 23% to 25%) lower mortality across between 17 and 69 years of follow-up (*Calvin et al., 2011*). In contrast, using molecular genetic data we found little evidence of a large effect of intelligence on mortality. This inconsistency may be because this sample of the UK Biobank was only followed-up for an average of 7.8 years, limiting our power to detect an effect on a relatively rare outcome (<2%), or because of selection bias. A large literature has investigated the associations of education and mortality and morbidity. For example, *Galama et al., 2018* reviewed studies which had used RCTs and natural experiments to estimate the effect of education on mortality, obesity and smoking (*Galama et al., 2018*). They found that estimates of the effect of educational attainment were heterogenous. Our results suggest that education has a larger direct effect on ever or current smoking. We found some evidence of a direct effect of education, but not intelligence, on BMI. The difference between our study and the studies reviewed in Galama et al. may be that we used a continuous rather than binary measure of adiposity. However, we still found evidence that education affected rates of overweight (67% prevalence) and obesity (24% prevalence) using the individual participant data analysis reported in *Supplementary file 1 - Figure 4*. *Gathmann et al. (2015)* reviewed the effect of 18 school reforms across Europe on mortality (*Gathmann et al., 2015*). They estimated that an additional year of schooling reduced male mortality over 20 and 30 years by 1.7% (95%CI: 0.2% to 3.2%) and 3.9% (95%CI: 1.8% to 6.0%). They report weak evidence that the effects on female mortality were smaller (0.9%, 95% CI: −1.2 to 2.9% and 1.8%, 95% CI: −0.1 to 3.6% reductions at 20

and 30 years respectively, p-value for difference by gender = 0.27 and 0.07). *Hamad et al. (2018)* systematically reviewed the literature using quasi-experimental methods to estimate the effects of education. They included 89 studies, and found that education had small beneficial effects on mortality, smoking and obesity (*Hamad et al., 2018*). Our results using genetic data for mortality were imprecise as the number of deaths to date has been relatively low. Over time as the number of deaths increase there will be more precision to investigate this hypothesis in the UK Biobank.

Researchers using univariable Mendelian randomization have found that education generally reduces risk of a range of outcomes (*Tillmann et al., 2017*). Larson et al. used two-sample Mendelian randomization and found that higher educational attainment reduced the risk of Alzheimer's disease. (*Larsson et al., 2017*) Potential explanations for these apparent protective effects of education on later health outcomes are: that they were due to (1) vertically pleiotropic effects of SNPs via intelligence and subsequently education (*Figure 1a*), (2) horizontally pleiotropic effects of the SNPs via intelligence (*Figure 1b*), and (3) confounding pleiotropic effects of the SNPs where the SNPs affect intelligence and education, but only intelligence affects the outcomes (*Figure 1c*). The total effects of intelligence and education on income had a similar direction. There was evidence of a total effect of education, but little evidence of a total effect of intelligence on many measures of morbidity and mortality and BMI (*Supplementary file 1 - Table 3*). The direct effects of education and intelligence, estimated by multivariable Mendelian randomization, were smaller than the total effects. The direct effects of intelligence were generally smaller than the direct effects of education (*Figure 4*). One explanation for this is that the SNPs have vertically pleiotropic effects via intelligence and education on the outcomes. The non-zero direct effects of education suggest that the effects of education estimated by univariable Mendelian randomization are unlikely to entirely be due to horizontally pleiotropic (direct) effects on the outcomes via intelligence. In another study, we used multivariable Mendelian randomization to estimate the direct effects of intelligence and educational attainment on Alzheimer's disease. The relationships between intelligence and education and Alzheimer's disease was largely due to an effect of intelligence (*Anderson et al., 2018*).

As with all analytic methods, inferences using multivariable Mendelian randomization depend on assumptions. Specifically: (1) the SNPs associate with intelligence even after conditioning on education and vice versa, (2) there are no confounders of the SNP-outcome associations, and (3) intelligence and/or education mediate all of the effects of the SNPs on the outcome (i.e. no horizontal pleiotropy mediated via factors other than intelligence or education). The first assumption is stronger than required for univariable Mendelian randomization (*Sanderson et al., 2018*). In the single sample setting, this can be tested using the Sanderson-Windmeijer test (*Sanderson and Windmeijer, 2015*). The SNPs exceed the critical values for this test for each exposure (*Supplementary file 1 - Table 4*) and the results are unlikely to suffer from weak instrument bias. It is not possible to prove the second assumption (independence) holds, because some confounders may be unmeasured or remain unknown.

This study investigated one of the most likely horizontally pleiotropic mechanisms in a univariable Mendelian randomization analysis of the effect of education on outcomes later in life: intelligence. SNPs which associate with intelligence in GWAS also strongly associate with education (*Okbay et al., 2016*). Indeed it would be surprising if there were SNPs that associated with intelligence that did not influence education. However, are these associations due to horizontal or vertical pleiotropy? Vertical pleiotropy would imply that the SNPs' effects on the outcomes are via an effect on intelligence and subsequently education. Horizontal pleiotropy would mean that the SNPs have direct effects on the outcomes via intelligence which are not mediated via an effect on education (*Figure 2*). Multivariable Mendelian randomization estimates of the direct effects of intelligence and education on the outcomes. Thus, they allow for any horizontal or vertical pleiotropic effects via intelligence or education. The estimated direct effects of intelligence on smoking, household income, BMI, alcohol consumption and sedentary behaviour were substantially attenuated compared to the total effects (*Supplementary file 1 - Figure 3*). This attenuation suggests that a substantial fraction of the total effects of intelligence on the outcomes were mediated via education.

What are the policy implications of these results? Economists and policymakers have been interested in the consequences of education for outcomes later in life. The length of education has increased in many countries around the world. If education affects later health and social outcomes, then this may result in improvements in public health. Our results suggest that education is likely to affect health, although these effects are smaller than indicated by a naïve Mendelian randomization

analysis that assumes the SNPs have no horizontally pleiotropic effects via intelligence. Each additional year of education (0.19 SD in UK Biobank) would result in a 2.0pp, 3.1pp, 2.4pp, 1.0pp increase in probability of having household income above £18 k, £31 k, £52 k and £100 k respectively, a 0.04 units increase in alcohol consumption (on a 1 to 5 Likert scale), 8.87 fewer minutes of sedentary behaviour per day, and 0.06 more days vigorous activity per week.

## Limitations

A potential source of bias in Mendelian randomization studies is sample overlap. If the same sample is used to detect the SNPs used as instruments as is used in Mendelian randomization the results can be biased towards the observed exposure-outcome association (*Burgess et al., 2016*). We minimised the possibility of this bias by restricting our main results to two entirely non-overlapping samples (see methods and supplementary materials). We used a restricted set of SNPs for intelligence detected without using UK Biobank data as a sensitivity analysis using individual-level data. The results from both approaches were consistent. Pleiotropy could explain our results if the SNPs we used as instruments directly affect the outcome through mechanisms other than intelligence and education, e.g. personality. However, a systematic review of studies investigating the effects of non-cognitive skills on academic and health outcomes later in childhood found evidence of modest effects on academic outcomes, and very few estimates of their associations with health outcomes later in life (*Smithers et al., 2018*). Furthermore, the published estimates were consistent with substantial small study and publication bias.

The measure of intelligence in UK Biobank is relatively crude: a 13 item verbal-numeric reasoning test. In a multivariable adjusted phenotypic analysis, this would cause measurement error on the exposure and attenuation of the coefficient on intelligence. However, Mendelian randomization is implemented here as a form of instrumental variable analysis, and therefore is less likely to be affected by measurement error on the exposures than conventional analyses (*Sargan, 1958*). Both intelligence and educational attainment had similar values of the Sanderson-Windmeijer F-statistic.

The intelligence GWAS we used to identify SNPs associated with intelligence was conducted in older adults. Intelligence is relatively stable across the life-course. For example, *Deary et al. (2004)* found that scores on the Moray House Test (a mental ability test) taken at age 11 and around age 77 were correlated (r = 0.66) (*Deary et al., 2004*). On average, SNPs associated with intelligence in adults had similar effects on intelligence in children ($r_g$ = 0.71) (*Hill et al., 2016b*). As a result our estimates of the effects of intelligence will partially reflect the effects of adult intelligence on the outcomes. Non-genetic quasi-experimental evidence suggests that length of schooling affects adult intelligence (*Ritchie and Tucker-Drob, 2018*). If adult intelligence affects the outcomes, then the estimated direct effect of education would be attenuated by any direct effects of education on the outcomes mediated via adult intelligence. Thus, our estimates of the direct effects of intelligence may be overestimates because they also include effects of adult intelligence. The effects of childhood intelligence could be estimated using SNPs identified in a GWAS of intelligence in children. However, currently available GWAS of childhood intelligence are considerably smaller than those available for adult intelligence.

Mendelian randomization studies using samples of unrelated individuals can be biased by bias due to population stratification and difference in ancestry across variants and selection and participation bias (*Haworth et al., 2019*; *Taylor et al., 2018*). We investigated this by adjusting for additional covariates, and bias component plots (*Davies et al., 2017*). These analyses suggested that while the genetic scores for intelligence and education associate with some measures of early life experience and place of birth, the bias induced in our estimates may be limited.

Another potential explanation for these results is dynastic effects which occur if parents' intelligence or education directly affects their offspring's outcomes. SNPs associated with education in GWAS also associate with parental education. Non-inherited parental alleles at these loci also associate with offspring's outcomes via their expression in parental phenotypes (*Kong et al., 2018*). Furthermore, substantial fractions of the GREML-SNP estimates of the heritability of education may be due to indirect effects of parents (*Young et al., 2018*). Thus, our estimates of the effect of intelligence and education are likely to attribute these parental effects to the offspring's characteristics. Similarly, parents do not mate randomly and assort on educational attainment. The associations of non-inherited alleles of SNPs known to associate with education provide evidence about the size of these effects. The association of offspring outcomes and the non-transmitted polygenic score for

education was 31% and 29% of the size of the association of the transmitted scores from fathers and mothers respectively. Similarly, the size of the association of the non-transmitted education polygenic score with a broad measure of health outcomes was 37% and 42% of the transmitted score for fathers and mothers respectively. These results suggest that some of the effects we report could be due to assortative mating, dynastic (genetic nurture) effects, which can cause bias in Mendelian randomization studies (*Hartwig et al., 2018*). We investigated this using the siblings in UK Biobank, but our results were underpowered. Future studies should investigate this further using within-family studies with larger samples (*Lawlor et al., 2017*; *DiPrete et al., 2018*; *Warrington et al., 2018*; *Brumpton et al., 2019*).

The effects we report may be specific to the time-period that the UK Biobank participants have lived through. For example, we find evidence of effects on smoking rates, particularly in single sample analysis. However, smoking rates have declined since the 1960s, 1970s, and 1980s, when the UK Biobank participants left school. Changes to education policies today, such as recent changes in the United Kingdom to mandate education, or part-time training and apprenticeships up to the age of 18, may not have the same impact as we report (*UK Government, 2018*). The UK Biobank is more educated than the general population, which could cause selection bias. However, we have previously shown that reweighting the sample to account for the under sampling of less educated people did little to affect the Mendelian randomization results (*Davies et al., 2018b*).

We have only investigated the effect of intelligence and education on a limited number of outcomes reported in UK Biobank. There are differences in morbidity and mortality by intelligence and education for a wide range of disease outcomes, and the conclusions we report here, that the direct effects of education are bigger than intelligence may not hold for other outcomes. Future studies should apply multivariable Mendelian randomization to summary data on outcomes to investigate this hypothesis as efficiently as possible (*Inoue and Solon, 2010*; *Angrist and Krueger, 1995*; *Pierce and Burgess, 2013*). Multivariable Mendelian randomization is a flexible approach for estimating and evaluating possible pleiotropic pathways. Future studies could exploit these methods to elucidate the mechanisms that mediate the effects of intelligence and education on outcomes later in life.

In summary, we found evidence from genetic association studies that both intelligence and education might affect health and social outcomes later in life. The direct effects for education are larger than for intelligence, suggesting that much of the effect of intelligence on outcomes later in life may be mediated via the effect of intelligence on education.

## Materials and methods

### Sample selection and sample overlap

The UK Biobank is a cohort study that recruited 503,317 people aged between 38 and 73 years old between 2006 and 2010 in 21 study centres across the UK. See *Supplementary file 1 - Figure 1* for an illustration of the inclusions and exclusion of samples into the study. UK Biobank received ethical approval from the Research Ethics Committee (REC reference for UK Biobank is 11/NW/0382).

Mendelian randomization estimates can be affected by weak instrument bias if overlapping samples are used to select SNPs associated with the exposures (intelligence and education) and the outcomes (*Burgess et al., 2016*). To minimise risk of this bias we estimated the SNP-outcome associations using a sample that excluded any participants who were included in the GWAS used to select the SNPs. The bivariate analysis estimated the SNP-outcome associations using participants that were not included in the intelligence GWAS reported by *Hill et al. (2019)* or the education GWAS reported by *Okbay et al. (2016)*. This excludes participants who took the verbal-numeric reasoning test or were in the UK Biobank interim release. Therefore, we used the remaining 124,661 participants to estimate the association between the SNPs and the outcomes of interest in the two-sample analysis.

For the single sample analysis (more details below), we constructed weighted allele scores for intelligence and education using the Sniekers GWAS which only included the UK Biobank interim release and the Okbay discovery sample (*Sniekers et al., 2017*). These GWAS are smaller than Hill GWAS, so detected fewer SNPs associated with intelligence at p-value$<5 \times 10^{-08}$. This fact means that the analyses using these SNPs are less precise than the two-sample analysis used in the primary

analysis. We used 77,882 participants who took the verbal-numeric reasoning test but were not included in the interim release. However, because the single sample analysis was restricted to individuals with the verbal-numeric reasoning scores, we can test whether the intelligence and education genetic scores sufficiently associate with intelligence and education. We used the Sanderson-Windmeijer F-test to test whether the polygenic scores explained sufficient variation in intelligence (education) conditional on education (intelligence).

## UK biobank sample selection

Over 9.2 million people were invited to take part. Of these, just over half a million people attended a study clinic and consented to take part in the study. These individuals provided blood samples and were genotyped. Full details of the genotyping are provided elsewhere (*Fry et al., 2017*). Of these, we excluded individuals who were related or were not white British ethnicity as indicated by genetic ancestry. This definition is relatively conservative, but it helps minimise the risk of bias due to population stratification. We further restricted the sample to individuals born in England to ensure that they experienced a similar education system. The two-sample analysis estimates the associations of the outcomes and the SNPs identified in Hill et al. and Okbay et al. using participants who did not take the verbal-numeric reasoning test. The single sample analysis estimates the effects of intelligence and education using allele scores defined using by the Okbay et al. and Sniekers et al. SNPs using participants who took the verbal-numeric reasoning test but were not in the UK Biobank interim release. Therefore, there is no overlap between the samples used to estimate the SNP-exposure associations and SNP-outcome associations.

## Phenotype definition

We used a broad measure of depression that the participant had seen a GP for nerves, anxiety, tension, or depression (ID:2090) at either the initial assessment centre visit or any repeat assessment centre visit, or a HES record indicating depression as a primary or secondary reason for admission (ID: 41202, 41204 and ICD-10 = F32, F33, F34, F38, F39) (*Howard et al., 2018*). The only participants who had measures of arterial stiffness were those who took the verbal-numeric reasoning test, therefore we excluded this outcome. More details of the phenotype definitions are provided elsewhere (*Davies et al., 2018c*).

We defined educational attainment using the same algorithm as the educational attainment GWAS (*Okbay et al., 2016*). Educational attainment was coded using the answer to touch-screen questions about qualifications. We assigned participants to their highest level of education reported at either assessment centre visit. Those with degrees were assigned to 20 years of education; NVQs, HND or HNC qualification were assigned to 19 years of education; other professional qualifications were assigned to 15 years; A-levels or AS levels were assigned to 13 years; GSCEs, O-levels or CSEs were assigned 10 years of education; and none of the above were assigned to 7 years. Previous studies have found that using self-reported age had little impact on estimates of the effect of education (*Sanderson et al., 2019*). We dropped individuals who stated prefer not to say or did not have a value for this question from the analysis. This measure of educational attainment was standardised to mean zero and variance one.

We defined intelligence using the 'verbal-numeric reasoning' score from the test taken at the baseline assessment centre visits. We replaced missing values of this test for the initial assessment centre visit with values taken at first repeat assessment visit (N = 15,404). The participants answered 13 logic questions within two minutes. We standardised this variable to mean zero and variance one.

## Covariates

All analyses included 40 principal components of genetic variation, sex, age, year and month of birth, and an interaction of month and year of birth and sex.

## Genotype quality control and selection

Full details of our genotype quality control pipeline are described elsewhere (*Mitchell et al., 2017*). In brief, we excluded participants who had mismatching sex, those with non XX or XY sex chromosomes, extreme heterozygosity or missingness. We limited the analysis to 11,554,957 SNPs on the HRC panel, of which we further limited to the 7,303,122 SNPs which were available in both the Hill

and Okbay GWAS. We selected independent SNPs ($r^2$ <0.01 within 10,000 kb) that were associated with either intelligence or education at p<5 × $10^{-08}$. Where there was more than one SNP in a region that associated with the trait at p<5 × $10^{-08}$ we selected the SNP with the lowest p-value. We then took the combined list of SNPs associated with either intelligence or education and repeated this process of clumping and selecting the SNP with the lowest p-value in the Okbay GWAS to create a list of SNPs for both traits that were in linkage equilibrium. See *Supplementary file 1 - Figure 2* for flowchart.

## Bidirectional effects of intelligence and education

We used bi-directional Mendelian randomization to investigate the direction of causation between intelligence and education. We estimated the effect of intelligence on education using the 181 lead SNPs from Hill et al. and data on educational attainment from participants of the UK Biobank who did not take the verbal-numeric reasoning test and were not in the interim release (*Hill et al., 2019*). These individuals were not included in the Hill et al. GWAS. We estimated the effects of education on intelligence using the 75 SNPs reported in the *Okbay et al. (2016)* discovery sample and samples with intelligence measures from UK Biobank. These samples do not overlap. We estimated the effects using two-sample summary data Mendelian randomization. The primary analysis used inverse variance weighted estimators. As sensitivity analyses, we used MR-Egger, weighted median, and weighted mode estimators to investigate whether pleiotropy biased the IVW estimates (*Bowden et al., 2016a*; *Hartwig et al., 2017*; *Bowden et al., 2016b*). We adjusted all the summary estimates for month and year of birth, sex, interactions of month and year of birth and sex and 40 principal components of genetic variation. We report estimates of the instrument strength and heterogeneity of the instrument-exposure association and the estimated effect of each of the exposures on the social and health outcomes across different SNPs.

## Univariable analysis

We estimated the total effect of intelligence and education on each of the health and social outcomes using univariable Mendelian randomization. We used 181 SNPs from *Hill et al. (2018)* and 75 SNPs from the discovery sample of *Okbay et al. (2016)* as instruments for intelligence and education respectively. Our primary analysis used IVW estimators which assume no directional pleiotropy. These estimates ignore possible pleiotropic or mediated effects via the other exposure: education and intelligence. We estimated the effect of each of the phenotypes using methods that were robust to other forms of pleiotropy using MR-Egger, weighted median and weighted mode estimators (*Bowden et al., 2016a*; *Hartwig et al., 2017*; *Bowden et al., 2016b*). These can obtain consistent estimates of the causal effect if the pleiotropic effects are independent of the effects on the exposure, or if the majority or most frequent variants are not pleiotropic.

## Multivariable Mendelian randomization analysis

We used multivariable Mendelian randomization to estimate the direct effects of intelligence and education on each of the health and social outcomes. This method has been described in detail elsewhere (*Sanderson et al., 2018*). In brief, this method is based on standard instrumental variable methods which allow for multiple exposures. Each exposure must be sufficiently explained by the set of instruments have an instrument that explains a sufficient proportion of the variation in the exposure, conditional on the other exposure. In the single sample, the SNPs can be correlated with each other (i.e. in linkage disequilibrium) and in the single or two sample case can also correlate with more than one exposure. The strength of association of the proposed instruments and the exposure can be tested using Sanderson-Windmeijer tests (*Sanderson and Windmeijer, 2015*). As described elsewhere it is possible to use these methods with summary data from two separate samples to estimate the SNP-exposure and SNP-outcome associations. The two-sample approach allows us to efficiently combine information from multiple studies, not all of which have measured intelligence, education and the outcomes (*Pierce and Burgess, 2013*). Hence we can integrate more data and have more precise estimates. We estimated the multivariable effects of intelligence and education using linear regression weighted for by one over the standard error of the SNP-outcome association.

## Sensitivity analyses

We conducted a series of sensitivity analyses to investigate how sensitive our results were to the methods we used.

### Individual level allele score approach

We investigated the sensitivity of our results by applying an individual allele score approach using individual participant data in the UK Biobank. We constructed two allele scores for intelligence using 16 SNPs from Sniekers et al. (that only used UK Biobank interim release data), and 75 SNPs from Okbay et al. (that did not use UK Biobank) and estimated the effect of intelligence and education using the Stata command ivreg2. The standard errors allow for clustering by month of birth. This method estimates the association of the allele scores and education in UK Biobank. Therefore, this method does not assume that the SNPs have the same effects on intelligence and education in different samples.

### Alternative clumping methods

In the primary analysis, we clumped the data by selecting and clumping SNPs using the lowest p-value reported in Okbay et al. This may mean that the instruments for education are stronger than the instruments for intelligence. We investigated this by selecting and clumping using the p-values from Hill et al.

### Larger education GWAS containing UKBB

*Lee et al. (2018)* published a large GWAS of educational attainment using data from up to 1.1 million individuals. In principle, this could provide more power and precision for our analysis. However, this GWAS used the entire sample from the UK Biobank, and so largely overlaps with the sample we used to estimate the SNP-outcome association. As a result, estimates using the Lee et al. GWAS are likely to suffer from weak instrument bias. We used 181 SNPs from Hill et al. and 486 SNPs from Lee et al. to estimate the multivariable Mendelian randomization using the summary methods described above. As expected, the results using the Lee et al. SNPs were more precise and attenuated towards the observational estimates.

### Investigating bias

We used covariate balance plots to investigate whether the bias from excluding a limited set of covariates would result in more bias in the multivariable adjusted or instrumental variable analysis (*Davies et al., 2017*). These tests compare the size of the associations of the exposure and the instrument with the covariates. We cannot directly compare the exposure-covariate to instrument-covariate association because a given size of association will result in a much larger bias in the instrumental variable estimator. Therefore, covariate balance tests account for the strength of the instrument in effect by estimating the ratio of the instrument-covariate and instrument-exposure associations (e.g. via a Wald estimator). This analysis results in two estimates called bias terms, an estimate of the bias if the covariate was excluded from a linear regression analysis and the equivalent bias term for if the covariate was omitted from the instrumental variable analysis. We calculate confidence intervals around both bias terms to indicate uncertainty. We plot the two associations on a forest plot allowing easy comparison of the bias terms. We assessed bias across a limited set of covariates that indicate early life environment, including measures of geography, birth weight, having been breast fed, had a mother who smoked in pregnancy, comparative body size and height age 10, whether parents where alive, and number of siblings. We investigated whether including these additional variables as covariates in the individual level single sample analysis affected the results. Finally, we investigated whether the results were due to the family environment by including a family fixed effect in the sample of siblings from the UK Biobank.

## Data and code availability

The cleaned analysis dataset will be uploaded to the UK Biobank archive. Please contact access@uk-biobank.ac.uk for further information. The analytic scripts used to clean the data and produce the results are available on GitHub (*Davies, 2019*; copy archived at https://github.com/elifesciences-publications/ukbiobank-intell-vs-ea).

## Acknowledgements

The Medical Research Council (MRC) and the University of Bristol support the MRC Integrative Epidemiology Unit [MC_UU_12013/1, MC_UU_12013/9, MC_UU_00011/1]. David Hill is supported by Age UK (Disconnected Mind grant). Ian Deary is supported by the Centre for Cognitive Ageing and Cognitive Epidemiology, which is funded by the Medical Research Council and the Biotechnology and Biological Sciences Research Council (MR/K026992/1). The Economics and Social Research Council (ESRC) support NMD via a Future Research Leaders grant [ES/N000757/1]. No funding body has influenced data collection, analysis or its interpretation. This publication is the work of the authors, who serve as the guarantors for the contents of this paper. This work was carried out using the computational facilities of the Advanced Computing Research Centre - http://www.bris.ac.uk/acrc/ and the Research Data Storage Facility of the University of Bristol - https://www.bristol.ac.uk/acrc/research-data-storage-facility/. This research was conducted using the UK Biobank Resource. The data used in this study can be accessed by contacting UK Biobank (www.ukbiobank.ac.uk). This analysis was approved by the UK Biobank access committee as part of project 8786. Consent was sought by UK Biobank as part of the recruitment process. Quality Control filtering of the UK Biobank data was conducted by R Mitchell, G Hemani, T Dudding, L Paternoster as described in the published protocol 10.5523/bris.3074krb6t2frj29yh2b03x3wxj.

## Additional information

### Competing interests

Neil Martin Davies: reports a grant for research unrelated to this work from the Global Research Awards for Nicotine Dependence (GRAND), an independent grant making body funded by Pfizer. The other authors declare that no competing interests exist.

### Funding

| Funder | Grant reference number | Author |
|---|---|---|
| Economic and Social Research Council | ES/N000757/1 | Neil Martin Davies |
| Medical Research Council | MC_UU_12013/1 | Neil Martin Davies<br>Emma L Anderson<br>Eleanor Sanderson<br>George Davey Smith |
| Medical Research Council | MC_UU_12013/9 | Neil Martin Davies<br>Emma L Anderson<br>Eleanor Sanderson<br>George Davey Smith |
| Medical Research Council | MC_UU_00011/1 | Neil Martin Davies<br>Emma L Anderson<br>Eleanor Sanderson<br>George Davey Smith |
| Medical Research Council | MR/K026992/1 | Ian J Deary |
| Age UK | Disconnected Mind Grant | W David Hill |

The funders had no role in study design, data collection and interpretation, or the decision to submit the work for publication.

### Author contributions

Neil Martin Davies, Conceptualization, Data curation, Formal analysis, Funding acquisition, Visualization, Writing—original draft, Project administration, Writing—review and editing; W David Hill, Data curation, Writing—review and editing; Emma L Anderson, Eleanor Sanderson, Methodology, Writing—review and editing; Ian J Deary, Data curation, Funding acquisition, Writing—review and editing; George Davey Smith, Conceptualization, Funding acquisition, Methodology, Writing—review and editing

## Author ORCIDs
Neil Martin Davies https://orcid.org/0000-0002-2460-0508
George Davey Smith http://orcid.org/0000-0002-1407-8314

## Ethics

Human subjects: The UK Biobank obtained informed consent from all participants. The UK Biobank received ethical approval from the Research Ethics Committee (REC reference for UK Biobank is 11/NW/0382). This analysis was approved by the UK Biobank access committee as part of project 8786. Consent was sought by UK Biobank as part of the recruitment process.

## Decision letter and Author response

Decision letter https://doi.org/10.7554/eLife.43990.013
Author response https://doi.org/10.7554/eLife.43990.014

## Additional files

### Supplementary files
• Supplementary file 1. Supplementary results and methods file.
DOI: https://doi.org/10.7554/eLife.43990.008

• Transparent reporting form
DOI: https://doi.org/10.7554/eLife.43990.009

### Data availability

All code used to produce the results in this study will be uploaded to GitHub (https://github.com/nmdavies/ukbiobank-intell-vs-ea; copy archived at https://github.com/elifesciences-publications/ukbiobank-intell-vs-ea). All data used in this study will be archived with the UK Biobank. Researchers who obtain the relevant permissions from the UK Biobank data access committee will be able to access the dataset (http://biobank.ndph.ox.ac.uk/crystal/dset.cgi?id=1762).

The following dataset was generated:

| Author(s) | Year | Dataset title | Dataset URL | Database and Identifier |
|---|---|---|---|---|
| Neil Martin Davies, W David Hill, Emma L Anderson, Eleanor Sanderson, Ian J Deary, George Davey Smith | 2018 | Data from: Multivariable two-sample Mendelian randomization estimates of the effects of intelligence and education on health | http://biobank.ndph.ox. ac.uk/crystal/dset.cgi? id=1762 | UK Biobank, 1762 |

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
