## [Decision Letter]

Thank you for submitting your article "The effects of intelligence and education on health, a bidirectional two-sample Mendelian randomization analysis" for consideration by *eLife*. Your article has been reviewed by two peer reviewers, and the evaluation has been overseen by a Reviewing Editor and Eduardo Franco as the Senior Editor. The following individual involved in review of your submission has agreed to reveal his identity: Stephen Burgess (Reviewer #1).

The reviewers have discussed the reviews with one another and the Reviewing Editor has drafted this decision to help you prepare a revised submission.

Summary:

This paper consists of a series of Mendelian Randomization (MR) studies related to intelligence, education, and later-life outcomes attempting to disentangle the complicated relationship between them. The analyses included in this study appear to be comprehensive, using many of the latest tools that have been developed. The choice of methods is generally appropriate, and the presentation of results is clear and measured. This work is interesting, and a great example of the potential of the multivariable MR approach to disentangle the effects of related risk factors. However, we have major concerns regarding the extensive use of causal language as there is some evidence that the key assumptions of MR may not hold in this application.

Essential revisions:

1) It is appreciated that the authors include the assumptions required for univariate and multivariate MR and the discussion of the limitations. However, the discussion of these assumptions and limitations need to be more complete and more central in the paper. MR analyses live or die to the degree that they satisfy the assumptions of the method. When this entire discussion is added to a section at the very end of the paper, it makes it very difficult as a reader to assess the credibility of the evidence being presented. Some further formal evaluation of whether assumptions hold could be presented. If the genetic risk score is as associated with confounders (notably socio-economic) as education itself, then the benefit of MR is less clear. (Note some association with confounders is expected, as education itself will influence socio-economic variables somewhat.) Perhaps a comparison with parental socio-economics would be helpful here – genetic variants should be less associated with parental socio-economics (if the MR assumptions are satisfied).

2) Related to the above point, the causal language throughout the paper is too strong. For example, in the limitations section, the authors acknowledge that dynastic effects and assortative mating could be driving "some of the effects [they] report." Indeed, the data is consistent with a model where the nurturing environment affects health but where education and intelligence have no effect at all. Despite this, there is unqualified causal language in the title, Abstract and throughout the manuscript. This should all be greatly softened.

3) Intelligence and education are difficult to define and measure – particularly intelligence. This is an intrinsic concern of any research involving intelligence, but it is particularly relevant for research that makes causal inferences and public policy recommendations (as per the final line of the Abstract). What would it look like to increase intelligence? What aspect of intelligence would we increase? There is also a methodological concern here – while the multivariable Mendelian randomization analyses are important for understanding questions of aetiology, the univariable Mendelian randomization analyses are more important for understanding the impact of public health interventions. As most interventions to increase intelligence would do so via education. Hence the total effects of intervention on intelligence and education should be as important as the direct effects. We would therefore encourage a co-equal presentation of these results.

4) The reviewers raised concerns about the use of multivariable Mendelian randomization for two variables that are so highly correlated. This is alleviated by the reasonably healthy Sanderson-Windmeijer F statistics, but particularly when the direct effects are in opposite directions (as for several examples in Figure 4 in Supplementary file 1), we worry slightly that this is just an artifact of including two highly correlated predictors in a regression model – due to chance variation, one will end up with a positive estimate and the other with a negative estimate. Having said that, it is reassuring that this pattern doesn't hold for all the outcomes. What is the correlation between genetic associations with education and genetic associations with intelligence?

5) Measurement error – while you are correct that standard univariable Mendelian randomization is not particularly influenced by measurement error, the same is not necessarily true for multivariable Mendelian randomization. As this is based on multivariable regression, it is possible for measurement error in genetic association estimates to lead to bias in any direction. It may be the case that the expected bias is low, but we are not aware of any theoretical or simulation work on this topic.

6) While we understand removing participants in the interim release for the analysis of genetic associations with intelligence/education, these participants could be included in analyses for other outcomes. We understand if you prefer to keep a consistent sample definition for comparability, but you may get improvement in power by using a wider sample for the gene-outcome associations. As a side point, in Figure 1 in Supplementary file 1, could you make clear where the two samples come from in the two-sample analyses? Currently there is only one box, but logically we'd expect there should be two.

7) It was difficult to identify the exact procedures followed in the paper at times. This was especially the case when they describe what data are used and which summary statistics are used. For example, the authors state, "The characteristics of 124,661 participants of UK Biobank who met our quality control and inclusion criteria for our primary analysis are described in Table 1." What is considered the "primary analysis" of this paper? The bidirectional EA vs. intelligence analysis, the univariate analysis, or the bivariate analyses? Or perhaps all three of them? Given that the authors also describe using published GWAS results, it is not clear how these UK Biobank participants are used. Just for the outcome phenotypes? Additionally, the descriptions of the Sanderson-Windmeijer test and the clumping procedure is ambiguous and hard to follow.

[Editors' note: further revisions were requested prior to acceptance, as described below.]

Thank you for resubmitting your work entitled "Intelligence, education and health, evidence from bidirectional two-sample Mendelian randomization" for further consideration at *eLife*. Your revised article has been favorably evaluated by Eduardo Franco as the Senior Editor, M Dawn Teare as the Reviewing Editor, and two reviewers.

The manuscript has been improved but there are some remaining issues that need to be addressed in a new revision round before acceptance, as outlined below:

The evidence that the IV assumptions may have been violated is rather underplayed and this needs to be more firmly acknowledged and stated within this manuscript either in the Discussion or in the limitations. One reviewer also highlights that measurement error can have an impact on Multivariable MR.

*Reviewer #1:*

I'm happy with how the authors have edited the paper. While the paper relies on assumptions, these are clearly laid out and considerable effort has been made to assess the assumptions.

*Reviewer #2:*

I appreciate the authors' substantial revisions in response to comments of the reviewers. The specific steps that were taken by the researchers is much more clear. I also appreciate the new analyses to test the assumptions of the method. I still have two substantive concerns, however.

1) While the authors describe the limitations of the methods they employ, they give a much more rosy interpretation of their sensitivity results than I think is merited. The two well-powered pieces of evidence the authors present related to the validity of the MR assumptions are the published studies on indirect effects of parents and the evidence that the polygenic scores are strongly associated with covariates related to childhood environment. Despite this, the authors make conclusions like "the impact of these associations on the final results may be small" and that "Assumption 2 [no confounders] is plausible because of the random inheritance of alleles at conception". My conclusion upon reading the analyses is that the assumptions of MR probably don't hold in this case and that the bias is potentially substantial. Do the authors disagree? If so, the authors should provide evidence for why they think this is the case or what evidence they have that the bias induced by violations of this assumptions is negligible. Otherwise, the authors should use more conservative language throughout about their sensitivity analyses and should minimally include a line in the Abstract about how they find evidence that the MR assumptions may not hold in this case.

2) I was not able to follow what was being done exactly based on the description of the bias component plots in the last paragraph of the subsection “Sensitivity analysis”, subsection “Investigating bias” of the Materials and methods, and the figure legends for Figures 6 and 7 in Supplementary file 1. A description of the procedure for producing this plot should be clarified.

---

## [Author Response]

Essential revisions:1) It is appreciated that the authors include the assumptions required for univariate and multivariate MR and the discussion of the limitations. However, the discussion of these assumptions and limitations need to be more complete and more central in the paper. MR analyses live or die to the degree that they satisfy the assumptions of the method. When this entire discussion is added to a section at the very end of the paper, it makes it very difficult as a reader to assess the credibility of the evidence being presented. Some further formal evaluation of whether assumptions hold could be presented. If the genetic risk score is as associated with confounders (notably socio-economic) as education itself, then the benefit of MR is less clear. (Note some association with confounders is expected, as education itself will influence socio-economic variables somewhat.) Perhaps a comparison with parental socio-economics would be helpful here – genetic variants should be less associated with parental socio-economics (if the MR assumptions are satisfied).

The vast majority of the measures in the UK Biobank indicate phenotypes in later life – there is very little information about early life conditions. For example, the UK Biobank did not include any measures of parental background. We have published quite extensively on the relationship between the education score and various confounders.

We investigated the association between the Sniekers (which is a subset of the Hill et al. data) and Okbay score and a limited number of covariates indicating the environment because we had very little power in the bivariate analysis and current bias assessment tools only allow for one exposure. The results of this analysis are shown in Supplementary file 1, Figures 6 and 7, for the Sniekers and Okbay scores respectively around the time of birth or early life using covariate balance plots.^1^ We had to use the Sniekers rather than the Hill because the latter used UK Biobank data from the full UK Biobank release, however it should be noted that the genetic correlation between the Hill et al. data and the Sniekers data is rg = 1 as shown in Figure 3 of Hill et al. We have focused on the univariate analysis. We describe the methodology behind the bias plots more completely in Davies et al., 2017.^1^ The bias plots estimate the relative bias caused by omitting a specific component from either the multivariable adjusted regression analysis or the instrumental variable Mendelian randomization analysis. These estimates adjust for the same set of covariates as the main analysis (PCs, sex, age, sex*age interactions). They account for the fact that a given association with a covariate will result in a much bigger bias in an instrumental variable analysis because the instrumental variable analysis only uses a small proportion of the variation in the exposure. What the plots show is that there is some evidence that on average individuals born in the south of the United Kingdom, those born in less deprived areas, or those born closer to London have higher values of the Sniekers allele score. However, the bias estimates for Index of Multiple Deprivation (IMD) are very similar for both the score and the intelligence measure in UK Biobank. Participants whose mother did not smoke in pregnancy were likely to have higher values of the Sniekers allele score. There was little evidence of associations with the other covariates. This is fairly modest evidence of bias. The education score was associated with the measures of geography, there was evidence it associated with measures of early life experience, such as breastfeeding.

How should these plots be interpreted?

There were similar associations with number of siblings as with measured education. These plots indicate that the education score does associate with early life socioeconomic measures. However, the extent of bias caused by these associations is unclear. The total bias caused by any of these potential confounders is the product of the association of the covariate with the score and the association with the outcome. We can get an indication of the total bias induced in the effect estimates by running a sensitivity analysis adjusting for these variables. Note these variables have missing values, which reduces the overall sample size and power. We reran the individual participant data additionally adjusting for these covariates and present the results in Figure 5 in Supplementary file 1. As a result of the missing values there are fewer observations and wider confidence intervals. However, there are few detectable differences between these adjusted results and the results presented in Figure 5 in Supplementary file 1.

Another way to control for family background is to run an analysis using sibling fixed effects. This is equivalent to running a difference model within siblings. Any factors that are invariant across siblings, e.g. ancestry or shared familial environmental factors, will be controlled for. A disadvantage of this approach is that there are only 38,000 siblings in UK Biobank and controlling for a family fixed effects substantially reduces statistical power. We found little evidence that our estimates differed across these models. However, this may reflect limited statistical power. However, the vast majority of estimates were larger than the in the within family analysis than the univariate IVW analysis. Interestingly, the implausible estimated effect of education on height attenuated entirely to the null in these models, suggesting that an effect of education in the offspring is unlikely to be driving these results. The results of this analysis are shown in Table 1 in Supplementary file 1. However, this analysis is underpowered. Therefore, we ideally need more data to evaluate this issue. We are in the process developing a consortium of family studies to investigate this issue with a much larger sample of siblings and trios.

We have included a new paragraph in the Results describing this analysis:

“We investigated whether the genetic variants associate with age at baseline. Without adjustment a one standard deviation increases in the Sniekers et al. and Okbay et al. scores were associated with 0.05 (95%CI: 0.0012 to 0.098, p-value=0.044) and 0.058 (95%CI: 0.020 to 0.096 p-value= 0.003) increases in age in years at baseline. […] Overall this suggests that while the there is some evidence that the scores associate with covariates, the impact of these associations on the final results may be small.”

We have included a new paragraph in the Discussion covering these sensitivity analyses:

“Mendelian randomization studies using samples of unrelated individuals can be biased by bias due to population stratification and difference in ancestry across variants and selection and participation bias (Haworth et al., 2018; Taylor et al., 2018. […] These analyses suggested that while the genetic scores for intelligence and education associate with some measures of early life experience and place of birth, the bias induced in our estimates may be limited.”

We have included a new section in the Materials and methods describing this analysis:

“Investigating bias: We used covariate balance plots to investigate whether the bias from excluding a limited set of covariates would result in more bias in the multivariable adjusted or instrumental variable analysis.^1^[…] Finally, we investigated whether the results were due to the family environment by including a family fixed effect in the sample of siblings from the UK Biobank.”

2) Related to the above point, the causal language throughout the paper is too strong. For example, in the limitations section, the authors acknowledge that dynastic effects and assortative mating could be driving "some of the effects [they] report." Indeed, the data is consistent with a model where the nurturing environment affects health but where education and intelligence have no effect at all. Despite this, there is unqualified causal language in the title, Abstract and throughout the manuscript. This should all be greatly softened.

The aim of our paper is to investigate the causal effects of intelligence and education on health outcomes. All Mendelian randomization estimates are dependent on assumptions, which we define in the Introduction. In the Discussion we describe possible sources of bias. However, throughout we refer to a) total effects and b) direct casual effects.

We have edited the manuscript as suggested. We have changed the title to: “Intelligence, education and health, a bidirectional two-sample Mendelian randomization analysis”.

We have edited the Abstract to read:

“Intelligence and education are both predictive of better physical and mental health, socioeconomic position (SEP), and longevity. […] If the assumptions of Mendelian randomization hold, these findings suggest a causal relationship with intelligence and education both being potential targets for interventions aimed at improving health outcomes.”

In the Introduction we have revised the start of the second paragraph to read:

“Mendelian randomization is an approach that can provide evidence about the relative causal effects of intelligence and education on social and health outcomes under specific assumptions.”

Throughout we have revised the manuscript to refer to estimated total effects and direct causal effects.

3) Intelligence and education are difficult to define and measure – particularly intelligence. This is an intrinsic concern of any research involving intelligence, but it is particularly relevant for research that makes causal inferences and public policy recommendations (as per the final line of the Abstract). What would it look like to increase intelligence? What aspect of intelligence would we increase? There is also a methodological concern here – while the multivariable Mendelian randomization analyses are important for understanding questions of aetiology, the univariable Mendelian randomization analyses are more important for understanding the impact of public health interventions. As most interventions to increase intelligence would do so via education. Hence the total effects of intervention on intelligence and education should be as important as the direct effects. We would therefore encourage a co-equal presentation of these results.

The test used to measure intelligence in the Hill et al. data set was primarily the verbal numerical reasoning test (VNR). This test has been shown to genetically correlate at 0.81 (S.E. = 0.09, p = 6.2 X 10^-18^) with a general factor of cognitive ability derived by extracting the first unrotated principal component from a battery of psychometric tests.^2^ General factors derived from different batteries of cognitive tests have very high phenotypic correlations, often in excess of 0.95,^3,4^ and are used in epidemiological studies, brain imaging studies, as well as genetic studies.^5^ As the Hill et al. intelligence phenotype is genetically highly similar to a general factor of cognitive ability, intelligence, whilst difficult to measure, has been measured in the current study in a manner consistent with that used in the literature.

As far as we are aware, with the exception of education, there are no interventions that can increase intelligence over the long term.^6^ Whether the association of education and later outcomes is due to intelligence, or education, is a key question for socioeconomic researchers – for exactly the reasons you give – the policy implications. If education associates with later outcomes because of pre-existing differences in intelligence between individuals, then policies to increase the length of schooling, or other educational interventions are unlikely to affect later outcomes. Conversely, if the effects of intelligence are mediated via schooling, then it suggests that both exposures can potentially be intervened on to affect outcomes. This study is relatively unusual because we have plausible natural experiments for both intelligence and education. While, as with all empirical analysis, these estimates depend on specific assumptions, they provide a new source of evidence about this debate. We have added the total effects to the Abstract.

4) The reviewers raised concerns about the use of multivariable Mendelian randomization for two variables that are so highly correlated. This is alleviated by the reasonably healthy Sanderson-Windmeijer F statistics, but particularly when the direct effects are in opposite directions (as for several examples in Figure 4 in Supplementary file 1), we worry slightly that this is just an artifact of including two highly correlated predictors in a regression model – due to chance variation, one will end up with a positive estimate and the other with a negative estimate. Having said that, it is reassuring that this pattern doesn't hold for all the outcomes. What is the correlation between genetic associations with education and genetic associations with intelligence?

If two exposures are very highly correlated, then multivariate Mendelian randomization will have lower power. There is a large literature on testing for weak instruments with multiple exposures, e.g. see references. This literature led to the development of multivariate weak instrument tests – such as the Sanderson-Windmeijer F-test. The null hypothesis of this test is that there is insufficient variation to identify the separate effects of the two exposures. This statistic essentially tests whether the instruments explain sufficient variation in one exposure, conditional on the other exposure.

Intelligence and education are very highly correlated – but not perfectly correlated. As a result, the F-statistic falls from 548 and 1661 for intelligence and education in the univariate case, to 27.2 and 27.4 for the multivariate Sanderson-Windmeijer F-statistic. This provides strong evidence that while there is far less statistical power for the multivariate analysis, the instruments do explain sufficient variation in the two exposures, and the analysis is unlikely to suffer from weak instrument bias.

Lee and colleagues estimated that the genome-wide genetic correlation between cognitive performance and educational attainment were genetic correlated r_g_=0.662. The coefficients on intelligence and education for the SNPs included in this study were correlated to a very similar degree r_g_=0.64 for the 194 SNPs selected from the Hill et al. and r_g_=0.59 for the 75 SNPs selected from the Okbay et al. GWAS. Thus, while most of these SNPs affect intelligence and educational attainment in the same direction, they do not have identical effects on each trait. We use this difference in the size of effect of the SNPs on each trait to identify the effects of intelligence and educational attainment.

5) Measurement error – while you are correct that standard univariable Mendelian randomization is not particularly influenced by measurement error, the same is not necessarily true for multivariable Mendelian randomization. As this is based on multivariable regression, it is possible for measurement error in genetic association estimates to lead to bias in any direction. It may be the case that the expected bias is low, but we are not aware of any theoretical or simulation work on this topic.

Ordinary least squares (OLS) will be biased towards the null if there is random measurement error on the exposures. Random measurement error on the exposure does not affect the covariance of the outcome and the exposure. Whereas random measurement error will inflate the variance of the measured exposure relative to the true underlying exposure. These terms are the numerator and denominator of the OLS estimator, and hence if the denominator of the estimator is inflated, the OSL estimate will be downwards biased. In contrast the instrumental variable estimator is the instrument-outcome covariance divided by the instrument-exposure covariance. If the measurement error is independent of the instrument, then it will not affect the instrument-exposure covariance.

We do not know of any theoretical reason why random measurement error on the exposures would affect multivariable instrumental variable estimates. We have conducted a simple simulation to investigate this point. We found no evidence that measurement error would cause bias to a multivariable instrumental variable estimate.

We simulated three random error terms for the outcome and two exposures, and a random confounder:

v,u,e,c~N0,1.

We generated correlated instruments:

rg~N0,1

zx1=0.3*N0,1+0.7*rg

zx2=0.3*N0,1+0.7*rg

We generated the exposures:

x1=zx1+c+v

x2=zx2+c+u

We generated the outcome:

y=x1+x2+c+e

We generated two miss measured exposures. Measurement error:

me1,me2~N0,1

Miss measured variables:

x1,measured=x1+me1

x2,measured=x2+me2

We ran instrumental variable regression (2SLS) for both measured and true underlying exposures. The true exposures were unbiased (β=1.01, SE=0.007 and 0.986, SE=0.007). When two miss measured exposures were included the estimates were unbiased (β=1.01, SE=0.012 and 0.977, SE=0.012). When one exposure was miss measured, these estimates were still unbiased, but more precise than using two miss measured exposures (β=1.01, SE=0.0084 and 0.982, SE=0.0084). In these simulations, random measurement error increases the standard error of the estimates, but it does not cause bias.

We have included the code for this simulation below.

//Neil Davies 12/03/2019

//This simulates a multivariable MR study with measurement error on the exposures

clear

set obs 250000

//Generate errors

gen v=rnormal()

gen u=rnormal()

gen e=rnormal()

gen c=rnormal()

//Generate correlated instruments

gen rg=rnormal()

gen z_x1=0.3*rnormal()+0.7*rg

gen z_x2=0.3*rnormal()+0.7*rg

//Generate the exposures

gen x1=z_x1+c+v

gen x2=z_x2+c+u

//Generate the outcome

gen y=x1+x2+c+e

//Run IV regression

ivreg2 y (x1 x2 =z*),ro

//Repeat with measurement error

//Generate measurement error

gen me1=rnormal()

gen me2=rnormal()

gen x1_measured=x1+me1

gen x2_measured=x2+me1

//Run IV regression

ivreg2 y (x1_measured x2_measured =z*),ro

//Remains unbiased

//Run IV regression with one miss measured and one measured accurately variable

ivreg2 y (x1 x2_measured =z*),ro

6) While we understand removing participants in the interim release for the analysis of genetic associations with intelligence/education, these participants could be included in analyses for other outcomes. We understand if you prefer to keep a consistent sample definition for comparability, but you may get improvement in power by using a wider sample for the gene-outcome associations. As a side point, in Figure 1 in Supplementary file 1, could you make clear where the two samples come from in the two-sample analyses? Currently there is only one box, but logically we'd expect there should be two.

The interim release of the UK Biobank was included in the Hill et al. intelligence GWAS used to identify the SNPs associated with intelligence. As a result, if the outcomes of these individuals were included in the single sample analysis it could induce bias. While we acknowledge that this bias may well be small, the additional power from including these individuals in the single sample analysis is also likely to be modest. Therefore, we would prefer not to include these individuals in the single sample analysis. We excluded these individuals from the two-sample analysis for the same reason. We have added a note to Figure 1 in Supplementary file 1 that the exposure data for the two-sample analysis comes from the Hill et al. and Okbay et al. papers.

7) It was difficult to identify the exact procedures followed in the paper at times. This was especially the case when they describe what data are used and which summary statistics are used. For example, the authors state, "The characteristics of 124,661 participants of UK Biobank who met our quality control and inclusion criteria for our primary analysis are described in Table 1." What is considered the "primary analysis" of this paper? The bidirectional EA vs. intelligence analysis, the univariate analysis, or the bivariate analyses? Or perhaps all three of them? Given that the authors also describe using published GWAS results, it is not clear how these UK Biobank participants are used. Just for the outcome phenotypes? Additionally, the descriptions of the Sanderson-Windmeijer test and the clumping procedure is ambiguous and hard to follow.

The primary analysis is the two-sample multivariable Mendelian randomization of the effects of intelligence and education on the outcomes, as this is the approach that has the smallest standard errors and makes most efficient use of the available data.

We have amended the Introduction to include:

“Our primary analysis uses two-sample multivariable Mendelian randomization of the effects of intelligence and education on a range of socioeconomic and health outcomes. This approach makes most efficient use of available data. We present single sample analysis as sensitivity analyses.”

We have edited the start of the Results section to make this and the how the different samples were used clear:

“The characteristics of 124,661 participants of UK Biobank who met our quality control and inclusion criteria for our primary two-sample multivariable analysis are described in Table 1. We take estimates of the SNP-intelligence and SNP-exposure associations from published GWAS of intelligence and education (Hill et al. and Okbay et al.), and estimate the SNP-outcome associations using participants of the UK Biobank who were not included in either GWAS.”

In the Materials and methods we have clarified how the clumping was conducted:

“We restricted the analysis to SNPs in linkage equilibrium which were identified in the intelligence and/or education GWAS at p<5x10^-08^ clumped on r2=0.01 within 10,000kb using the 1000 genomes reference panel.^7^[…] For these pairs of SNPs, we selected the SNP that most strongly associated with education in the GWAS.”

We have added a further explanation of the Sanderson-Windmeijer F-statistics. Essentially these are just tests of whether the SNP explains the exposure after conditioning on the other exposure:

“Sanderson-Windmeijer F-statistics tests the strength of the SNP-exposure conditional on the other exposure (intelligence or education). Because the effects of the SNPs on intelligence and education are similar (but not identical), the Sanderson-Windmeijer multivariable F-statistics are smaller than standard univariable F-statistics.”

Please let us know if there are any other aspects of the paper we can clarify.

References:

1) Davies, N. M. et al. How to compare instrumental variable and conventional regression analyses using negative controls and bias plots. Int J Epidemiol (2017). doi:10.1093/ije/dyx014

2) The neuroCHARGE Cognitive Working group et al. Molecular genetic aetiology of general cognitive function is enriched in evolutionarily conserved regions. Translational Psychiatry 6, e980–e980 (2016).

3) Johnson, W., Bouchard, T. J., Krueger, R. F., McGue, M. and Gottesman, I. I. Just one g: consistent results from three test batteries. Intelligence 32, 95–107 (2004).

4) Johnson, W., Nijenhuis, J. te and Bouchard, T. J. Still just 1 g: Consistent results from five test batteries. Intelligence 36, 81–95 (2008).

5) Deary, I. J. Intelligence. Annu Rev Psychol 63, 453–482 (2012).

6) Ritchie, S. J. and Tucker-Drob, E. M. How Much Does Education Improve Intelligence? A Meta-Analysis. Psychological Science 29, 1358–1369 (2018).

7) Hemani, G. et al. The MR-Base platform supports systematic causal inference across the human phenome. eLife 7, (2018).

[Editors' note: further revisions were requested prior to acceptance, as described below.]

Reviewer #2:

I appreciate the authors' substantial revisions in response to comments of the reviewers. The specific steps that were taken by the researchers is much more clear. I also appreciate the new analyses to test the assumptions of the method. I still have two substantive concerns, however.1) While the authors describe the limitations of the methods they employ, they give a much more rosy interpretation of their sensitivity results than I think is merited. The two well-powered pieces of evidence the authors present related to the validity of the MR assumptions are the published studies on indirect effects of parents and the evidence that the polygenic scores are strongly associated with covariates related to childhood environment. Despite this, the authors make conclusions like "the impact of these associations on the final results may be small" and that "Assumption 2 [no confounders] is plausible because of the random inheritance of alleles at conception". My conclusion upon reading the analyses is that the assumptions of MR probably don't hold in this case and that the bias is potentially substantial. Do the authors disagree? If so, the authors should provide evidence for why they think this is the case or what evidence they have that the bias induced by violations of this assumptions is negligible. Otherwise, the authors should use more conservative language throughout about their sensitivity analyses and should minimally include a line in the Abstract about how they find evidence that the MR assumptions may not hold in this case.

There is evidence, both in our paper, and elsewhere (e.g. Kong et al., 2018, Haworth et al., 2019, Abdellaoui et al., 2018) that the education associated variants associate with family background. These associations can be induced by residual population stratification, assortative mating, and dynastic effects. However, these effects do not necessarily induce bias in Mendelian randomization estimates of the effect of education on other outcomes. The overall bias is a function of the association of the SNPs and family background, and the association of family background and the outcome. Any effect of the family background on the exposure (education) will not cause bias in the Mendelian randomization estimates. We assessed the sensitivity of our results to these sources of bias by comparing our results with and without adjustment for principal components. There was very little evidence that the effects differed. This is consistent with the principal components (which indicate geographic ancestry) not being a major confounder of the SNP-outcome associations. The second sensitivity analysis we conducted was the sibling analysis which perfectly controls for population structure, common dynastic effects, and assortative mating. Again, using this analysis we found little evidence that our estimates differed. Therefore, while the Mendelian randomization assumptions are unlikely to perfectly hold, we found little evidence that violations in these assumptions resulted in bias in our estimates. Finally, in previous papers, we have used the raising of the school leaving in 1957 to provide further evidence about the effects of education. We found consistent evidence of effects of education for a number of outcomes (Davies et al., 2018). Nevertheless, many of these sensitivity analyses are underpowered, and they are certainly insufficient to prove that the violations of the Mendelian randomization assumptions do not meaningfully affect or explain our results. We need more evidence, particularly from family studies to resolve this.

We have made the following edits:

Subsection “Limitations”: Assumption 2 is plausible because of the random inheritance of alleles at conception. It is not possible to prove the second assumption (independence) holds, because some confounders may be unmeasured or remain unknown.

Overall this suggests that while the there is some evidence that the scores associate with covariates, the impact of these associations on the final results may be small there is little evidence from these sensitivity analyses that these violations of the Mendelian randomization assumptions affects our results.

2) I was not able to follow what was being done exactly based on the description of the bias component plots in the last paragraph of the subsection “Sensitivity analysis”, subsection “Investigating bias” of the Materials and methods, and the figure legends for Figures 6 and 7 in Supplementary file 1. A description of the procedure for producing this plot should be clarified.

We have added the following sentence to the Results:

“This approach compares the omitted variable bias that occurs if a specific measured covariate is omitted from the multivariable adjusted and instrumental variable regression.”

We have added the following description to the Materials and methods:

“These tests compare the size of the associations of the exposure and the instrument with the covariates. […] We calculate confidence intervals around both bias terms to indicate uncertainty.”